# Manipulation and Localized Deposition of Particle Groups with Modulated Electric Fields

**DOI:** 10.3390/mi11020226

**Published:** 2020-02-23

**Authors:** David Pritchet, Kornel Ehmann, Jian Cao, Jiaxing Huang

**Affiliations:** 1Mechanical Engineering Department, Northwestern University, Evanston, IL 60208, USA; 2Materials Science and Engineering Department, Northwestern University, Evanston, IL 60208, USA

**Keywords:** process control, process characterization, dielectrophoresis, electrophoretic deposition, self-assembly, finite element analysis, electrophoretically-guided micro additive manufacturing (EPμAM)

## Abstract

This paper presents a new micro additive manufacturing process and initial characterization of its capabilities. The process uses modulated electric fields to manipulate and deposit particles from colloidal solution in a contactless way and is named electrophoretically-guided micro additive manufacturing (EPμAM). The inherent flexibility and reconfigurability of the EPμAM process stems from electrode array as an actuator use, which avoids common issues of controlling particle deposition with templates or masks (e.g., fixed template geometry, post-process removal of masks, and unstable particle trapping). The EPμAM hardware testbed is presented alongside with implemented control methodology and developed process characterization workflow. Additionally, a streamlined two-dimensional (2D) finite element model (FEM) of the EPμAM process is used to compute electric field distribution generated by the electrode array and to predict the final deposition location of particles. Simple particle manipulation experiments indicate proof-of-principle capabilities of the process. Experiments where particle concentration and electric current strength were varied demonstrate the stability of the process. Advanced manipulation experiments demonstrate interelectrode deposition and particle group shaping capabilities where high, length-to-width, aspect ratio deposits were obtained. The experimental and FEM results were compared and analyzed; observed process limitations are discussed and followed by a comprehensive list of possible future steps.

## 1. Introduction

The ability to move, order and deposit particles from colloidal solutions via modulated electric fields constitutes an essential technological platform for the development of new nanomaterials, functional surfaces, and enables studies of phase transitions via colloidal mockup models and characterizations of biological and nonbiological particulates. Xia et al. [1] presented a research overview of monodispersed colloidal spheres and their use as two-dimensional (2D) fabrication templates and as lithographic masks. The authors pointed out that the industrial application of these materials depended on their capability to form complex crystal structures (as compared with ccp, bcp, and rhcp 2D crystal lattices) with minimized defect densities. Achieving the precise geometry requirements for colloidal structures necessitates large tolerance for internal defects, which can be detrimental to the part function. Russel [2] presented a tunable colloidal system model used to study the phase transitions and crystal nucleation. The tunability of the colloidal system, based on the adjustment of particle volume fraction and the amplitude of the applied electric field, demonstrated the capability of the system to exhibit the complex, reversible phenomena and proved to be instrumental in discovering new phase transitions for colloidal crystals. This colloidal tunability via electric fields could also be employed to minimize, or completely fix, the internal defects in the colloid structure; however, it required in-depth knowledge of particle-field interactions during the self-assembly processes. In contrast to the granular nature of colloidal particles, the effects of electric fields on the mechanical deformation (i.e., creep) of aluminum samples were experimentally studied in [3], and via transmission diffraction electron microscopy in [4]. This approach could be adapted for experimental study and analysis of post-processed colloidal particle deposit, with a goal to elucidate the electric field capability to form compact parts and, alternatively, to modify the deposit structure. Juarez and Bevan [5] studied via numerical simulations and experiments particle colloidal (self)assembly and particle–field interactions. They used inverse Monte Carlo analysis to reconstruct colloid configurations from modeled interaction potential with particle-particle, particle-surface, and particle-field terms. The obtained good matches of the particle density and distribution profiles demonstrated the capability of their approach to obtain particle, medium, and electric field interaction parameters. In their topical review, Li et al. [6] mentioned that removal of particle surface stabilizers, during the post assembly stages, can inadvertently change the final composition, or even the shape of the deposits. The authors also mentioned that application of top-down methods (e.g., the use of masks and templates) could be limiting factor in the achievable pattern resolution and that emerging processes that used guided assembly via external force application (i.e., the bottom-up processes) could create 2D meta-stable deposits, in a thermodynamic sense, but with better pattern resolution. These material forming approaches rely on contactless control of the energy of the colloidal system and “activating” the material into a desired configuration. The concept of “active matter”, as explained by Ouellette in [7], and its perceived analogy to the behavior of animal groups (e.g., a flock of birds or school of fish), raises a question of adequacy for currently available characterization tools and fundamental models to sufficiently explain the observed material behavior. The author suggested building a rule-based model for the system in question and, then, attempting to infer physics of emergent material properties by comparing the a priori model output data with the experimental findings. This approach seems promising, especially for the study and characterization of colloidal particle systems and their responses to the applied external forces, notwithstanding the analogy with the animal group behavior.

In this article, we present a contactless method that is capable of local particle deposition and manipulation via application of modulated electric fields. We have named this method electrophoretically-guided micro additive manufacturing (EPμAM) and its schematic is shown in Figure 1a. The basis of this method is the electrophoretic deposition process (EDP), an industrial coating technique capable of producing uniform coatings on conductive surfaces [8] (Figure 1b). By modifying the EDP to use nonuniform electric fields, applied with an electrode array, one can exploit the dielectrophoresis (DEP) phenomenon to manipulate colloidal particles at greater precision as compared with electrophoresis-based particle ordering (Figure 1c). The dielectrophoresis technique [9] is commonly employed in biomedical sciences, where suspended (bio)particle separation and isolation is achieved by careful application of nonuniform electric fields with microelectrodes. The DEP forces are suitable for fine manipulation of particulate matter but usually have to rely on other types of forces that initiate and sustain the particle transport.

The EDP relies on the strength of the electric field to deposit charged particles [10], whereas the DEP exhibits electric field frequency-dependent behavior, defined though material-dependent Clausius–Mossotti factor and acts along the direction of the electric field gradient [11]. Combining EDP and DEP capabilities on a single platform and with a suitable control mechanism would allow the deposition of non-charged particles, a limitation of classical EDP. Moreover, for the same electric field, the DEP forces will always be one magnitude smaller/finer than the accompanying EDP forces. This stems from DEP force dependence on the electric field gradient as compared with the EDP force dependence on the electric field strength (i.e., field magnitude). By utilizing the frequency control of the magnitude of the DEP forces, via the DEP force dependence on the Clausius–Mossotti factor values, finer control of the EDP can be achieved.

Use of uniform microelectrode and electrode arrays, instead of a complex electrode shape geometry, provides flexibility where almost arbitrary electric fields can be generated and controlled during the manipulation process. A notable example of such a device is a micro total analysis system, built with CMOS technology [12]. The system operated by creating so-called electric field cages that trapped and manipulated biological cells in a controllable manner. Another example of accurate manipulation of cells used projected images to create nonuniform electric fields and via DEP attract particles to the desired location [13]. A similar device was used to control the chemical concentration locally [14]. All these devices used digital control methodology to generate nonuniform electric fields at desired locations that attracted particle or a smaller number of particles. They also employed a (bi)planar electrode configuration that severely limited their workspace.

In our method, the applied electric fields are used to push particles, thus, allowing for larger workspace and better reusability of the electrode array because it is not necessary to remove the deposit from the electrode; our setup can locally deposit particle groups away from the electrodes, before depositing a new layer, or a group of particles. Moreover, by pushing particles and depositing them between the electrodes, the EPμAM process essentially decouples the resolution requirement of the desired geometrical feature from the actual tool (e.g., microelectrode or deposition mask) that needs to produce and build it. In this way, the use of microelectrode array, coupled with a flexible control system and the developed FEM process models, allows the EPμAM process to scale up or down the electrode size easily (and consequently the deposition area) without the need for costly, and complex, retooling device modifications.

This article is organized as follows: In the Methods section, we present our experimental testbed, show the developed empirical workflow, illustrate the implemented control scheme, and numerical simulations of the simplified EPμAM process. The Results section contains basic process characterization and advanced manipulation experiments, performed with our experimental setup. These particle group manipulation experiments demonstrate our system’s capability to group, disperse, and move particle groups, comparable to the previously reported systems [12,13,14]. The performed advanced manipulation experiments present deposition between the electrodes, as well as shape manipulation capabilities for particle groups. In the Discussion section, we comment on the obtained deposit quality and discuss observations and process limitations. In Conclusions, we offer a few possible directions for further development and application of the EPμAM process and the employed methodologies.

## 2. Methods

In this section, our experimental testbed, the EPμAM process characterization workflow, implemented control schemes, and numerical simulations of the simplified EPμAM process are presented. The testbed is designed to be highly modular and reconfigurable which minimizes instrument downtime when electrode arrays need to be replaced. The implemented workflow and the control schemes are devised to facilitate fast process characterization. The developed multiphysics FEM model of EPμAM deposition process is used for prediction of final particle deposit locations and is intended to be used as a platform for modeling physical phenomena relevant to the EPμAM process.

### 2.1. Experimental Setup

The schematic of the experimental setup is shown in Figure 2. The testbed that holds deposition cell and the inverted microscopy setup is located inside a lightproof enclosure, designed to remove environmental light and minimize the light noise in the captured images.

The microscopy setup consisted of a USB camera (Edmund Optics USB color camera EO1312C), extension tube, tube lens (Edmund Optics MT-4), and long working distance objective (2X Mitutoyo Plan Apo Infinity Corrected #46-142). The USB camera was connected to the EPμAM workstation (Dell Precision T5810 Workstation, running 64bit Windows 7) and allowed programmable capture of image frames (i.e., image snapping). The same EPμAM workstation controls the relay switchboard via MATLAB code. The relay switchboard (NCD Industrial Relay Controller 16-Channel SPDT) modulated the electric power received from the variable DC power supply (Electro Industries Inc. Model 4045) and applied an electric potential to the microelectrode array. The microelectrode array had 16 electrodes (Figure 3) and was located on the top of the deposition cell, which allowed for an unobstructed view of particle movement and grouping with the inverted microscope.

The microelectrode array, shown in Figure 3, was in-house built with header pins (SamIdea 2.54 mm pitch male to female header), a custom microelectrode array holder, and AWG 22 (0.6438 mm wire diameter) solid core wiring. The electrode pin geometry selection was based on our previous work on electrode geometry shape optimization in the context of the EPμAM process [15]. The use of a custom microelectrode array holder was necessary to ensure that no crosstalk occurred between the microelectrode’s 16 channels. The left side of Figure 3 shows the CAD assembly of microelectrode array, microelectrode holder, and the kinematic mount. The kinematic mount was used to align and level the microelectrode array and to attach it to the EPμAM testbed. The assembled microelectrode array detail is shown on the right side of Figure 3. A piece of adhesive tape was placed over the bottom part of the array in order to minimize reflection of the array’s metallic parts. This greatly improved the signal-to-noise ratio of the microscope’s images.

The developed control system in MATLAB was based on our previous work [15,16]. We generalized the control algorithm that produced the pulse trains. Now, the control algorithm controls the relay’s switches and has explicit control over the pulse-width modulated (PWM) duty cycles. This control setup allows us to achieve up to 4 Hz of pulse train frequency on the high end and arbitrarily low frequency on the low end of the control frequency range when the high current power supply was used (i.e., when working current was higher than 0.01 A). Examples of pulses with 100%, 75%, and 50% duty cycle are shown in Figure 4. The amplitude of the applied electric field was the same, and the PWM technique regulated the average potential value with the duty cycle variation. The pulses have negligible rising times.

### 2.2. EPμAM Process Characterization Workflow

The workflow for the characterization experiments is shown in Figure 5. The upper right corner of Figure 5 shows a schematic of the used deposition cell and accompanying naming convention for its parts. The reader is referred to this naming convention for interpretation of the results in Section 3.1 of this article. The workflow consisted of a sequence of experiments performed on a single sample, where particle movement and grouping were observed via an inverted microscope setup. This movement and grouping were induced by the application of electric potential during each actuation cycle. The electric potential application was defined with the devised control schemes (shown in Figure 6).

The design of this workflow allows for fast testing of various materials, without the need to significantly change the approach for different particle and fluid systems. In this work, we are focusing on two parameters, particle concentration and the strength of the applied electric current. The particle concentration parameter is important because it represents the primary input to the EPμAM process– an amount of particulate material. Depending on the particle size and its creation process, the amount of used particulate material can be the main economic factor for the decision to use EPμAM process to produce parts. The electric current strength represents a convenient proxy for EPμAM’s testbed capability to manipulate particles. Using higher or lower currents dictates the necessary thickness of the electrical wiring, durability of the microelectrodes, and power and safety requirements for the used switching circuits. For the initial characterization attempt of the EPμAM process, we will be solely focusing on these two parameters.

### 2.3. Employed Control Schemes

During each actuation cycle, the 16-channel relay switchboard (SPDT switch type) connects each electrode to the power supply’s either high or low voltage. In our experiments, high voltage was always set to 15 VDC, and the low voltage was set to 0 DC (i.e., ground). Examples of the used control schemes are shown in Figure 6. Each control scheme had two states, and each state duration depended on the set duty cycle. For example, in Figure 6a, all outer cells were initially set to the high voltage (red cell with number 15 in it), and all inner cells were set to the low voltage (blue cell with number 0). This actuation configuration was defined as State 1. Each cell defined an electric potential value for a single electrode in the electrode array. State 1 was held for Δt_1_ time, after which it was switched to State 2, where all cells were set to low voltage. State 2 was held for Δt_2_ time and, then, switched back to State 1. The sum of Δt_1_ and Δt_2_ corresponded to a single actuation period, i.e., cycle. Control schemes used for simple actuation strategies (control schemes A–D, shown in Figure 6a–d) had Δt_1_ and Δt_2_ set to 0.75 s and 0.25 s, respectively. Advanced control schemes, used for midline particle deposition (control schemes E and F, Figure 6e,f), had both Δ*t*_1_ and Δ*t*_2_ set to 0.5 s.

The control scheme in Figure 6a was used to group particles in the middle of the electrode array, between the inner four electrodes. Figure 6b shows the control scheme used to disperse particles from the middle of the electrode array and deposit them on the edges of the array. The control scheme used to move particle groups to the third column of the array and deposit them near these electrodes is shown in Figure 6c. The control scheme used to move particles to the right edge of the array and deposit them near the fourth electrode array column is shown in Figure 6d. The more advanced control scheme, used to “squeeze” (i.e., move and group) particles between the second and third rows of the electrode array, is presented in Figure 6e. Another advanced control scheme, used to stretch particle group between the first and the fourth array columns, is shown in Figure 6f. The control schemes, shown in Figure 6e,f, were used in sequence to obtain long, midline particle deposits, as described in Section 3.2 of this paper.

Examples of main control loop timing periods for executed 100 cycles are shown in Figure 7. In Figure 7a, the results of the main control loop timing for four control schemes are shown. It can be seen from Figure 7a that the first few cycles last longer due to the initialization of the communication channel between the PC workstation and the relay controller. Figure 7b presents box plots of data in Figure 7a. Due to communication latency between the PC workstation and the relay switchboard, a single actuation period lasted, on average 1.235 s, for the commanded cycle of 1 s.

### 2.4. FEM Model for EPμAM Process

The numerical simulations in COMSOL Multiphysics software (v5.3) were developed to compute the electric field distribution for given electrode signals and, then, calculate the test particle trajectories. The FEM model is provided in the online Appendix A. These simulations allowed us to double-check the computed switch relay commands and determine the expected electric potential distribution during the empirical tests. The simulations also allowed us to predict final particle locations, given the computed electric potential distribution. The generated 2D mesh and geometrical parameters used to build it are shown in Figure 8. The computation domain represented a top-down view of the deposition cell (see Figure 2 deposition cell detail, bottom view or upper right corner of Figure 5 for reference) and allowed for easy visual comparison between the experimental and the simulation results. Table 1 contains all physical and numerical parameters for the developed EPμAM process FEM model. Other FEM model parameters were left on their default settings.

The FEM model has two computational steps. In the first step, electric potential distribution was computed with the Electrostatics COMSOL module using a frequency study step. The boundary conditions that represent actuating electric potential were applied on the electrode’s surfaces (the small 16 circles’ circumferences in the center of Figure 8). Then, the computed electric potential from Step 1 was used as an input in Step 2 of the model. In the second step, which used particle tracing for the fluid flow COMSOL module in a time-dependent study step, the test particle group was released from the grid, and their trajectories were, then, computed for 110 s each 0.5 s, resulting in 220 timesteps. The model domain was assumed to be filled with water, and the particles have physical properties of rutile titania.

## 3. Results

The Results section contains experimental results from three performed studies. In the first study, simple particle group manipulation is demonstrated and compared to the FEM results. The experimental runs were repeated with varying particle concentration and electric current density. Particle group locations and distributions from these characterization experiments were quantitively compared by computing the binary mask area of deposited particles and plotting them versus the experiment time. In the second study, an advanced manipulation technique was presented where particles were deposited between the electrodes. In the third study, a particle deposit was prepared for characterization and quantitative evaluation on a laser confocal microscopy instrument. These preliminary results demonstrated the EPμAM process capability to organize and order particles by pushing them into the desired location.

### 3.1. Simple Particle Group Manipulation and Process Characterization

Preliminary deposition runs were designed to test the microelectrode array’s capability to move and deposit particle groups. At the beginning of each experimental run, a water solution (2% and 0.2% wt. solutions) of rutile titania (TiO_2_) particles were injected (100 to 200 μL injection volume) in the deposition cell with a syringe. The mini well was previously filled with 3 mL of water. The particles were left to diffuse and fall on the bottom of the cell before the electric fields were applied to them. This preparation stage usually lasted 10 to 20 s.

No detergents or chemical stabilizers were used for the preparation of the titania aqueous solution. Particle size measurements were performed with a Horiba LA-960 Laser Particle Size Analyzer, where no sonication, 1 min sonication, and 2 min sonication were applied to the sample prior to the measurement. These measurements were performed to check the colloidal stability. The obtained particle size distributions are shown in Figure 9. The sample sonication preparation shows that the used particle-fluid system is stable under mechanical agitation and has an average particle size between 1 and 2 μm.

After the injected particles were settled on the bottom of the deposition cell, the manipulation tests were performed in the following sequence, as shown in Figure 5: (i) Particles are focused in the middle of the array; (ii) then, particles are defocused from the middle of the array and grouped on the array’s edges; (iii) particles are translated to the third electrode array column (see upper right panel in Figure 5); and (iv) particles are translated to the edge of the array, to the fourth array column. Each corresponding experimental step consisted of the application of 100 cycles of the appropriate control schemes, as shown in Figure 5. During the particle movement, images were taken with the inverted microscopy setup every 10th cycle, which corresponded to every ~12 s of the process. In total, the whole experimental run, including the preparation stage, lasted for 500 to 510 s.

Experimental results for 2% wt. particle solution are shown in Figure 10. Figure 10a shows particle group after they have settled at the end of the preparation stage. Panels b–e in Figure 10 show particle group distribution at the end of the 100th cycle for each experimental run (i–iv). In the bottom right corner of all images is the 1 mm white length bar. Figure 11 shows the FEM model computed electric potential distribution for the control schemes A–D. Figure 12 shows the initial and end test particle positions for the computed electric potential distribution cases given in Figure 11.

Figure 10b depicts the final distribution of the focused particle group, delineated with the dashed gray rectangle in the middle of the electrode array. There are a few large particle groups left outside this deposition region, and they are marked with a yellow arrow (the readers are referred to the color version of Figure 10). These large groups formed agglomerations on the bottom of the mini well, and they were sufficiently large to enable particle packing which prevented particle conveyance with the employed electric fields. These groups were broken up by subsequent application of electric fields in reverse, indicating that these groups formed semistable deposits outside of the desired deposition region. Smaller particle groups which ended outside the desired deposition region are indicated with the red arrows. These particles did not seem to be agglomerated and settled, but just smeared over the mini well’s surface and they were easily moved by applying additional cycles of the same electric field distribution. Detail A in Figure 10a shows one notable particle formation, in the form of a deposited arch, that provided a good match with the obtained shape of the electric field distribution (Detail A, Figure 11a) and final positions of the test particles (Detail A, Figure 12b).

The obtained particle distribution after 100 application cycles of control scheme B is shown in Figure 10c. The majority of the particles are located within the annulus of the desired deposition region (dashed gray annulus in Figure 10c). It is worth noting that most particles within this region are not agglomerated (i.e., like large particle groups identified with yellow arrow), but lightly deposited, and with a form closer to the small particle groups, which are smeared (and identified with a red arrow in these figures). There are a few large groups of particles outside the desired deposition region and identified with a yellow arrow. Most likely, the application of electric potential caused them to overshoot and get latched to the electrodes (see, for example, Detail C in Figure 10c). After latching, these particles started to behave like an extension of the electrode in contact and started to form undesired particle agglomerations. Some particle groups are left in the middle of the annulus (the hole in desired deposition region in Figure 10c that encompasses the middle four electrodes), which was not a desired deposition region (see Detail B, Figure 10c). These particles are left in the middle of the array due to the locally formed electric potential minima (created within the middle four electrodes) that prevented them from migrating away.

The end particle distribution obtained with control scheme C is shown in Figure 10d. The majority of the agglomerated particles are located within the desired deposition region. Detail D in Figure 10d shows formation of a long deposit of agglomerated particles, with high length-to-width aspect ratio that spans between the third and fourth rows of the third electrode array column (see upper right panel in Figure 5 for definitions of these terms) that can be desirable in microelectronics (i.e., to form electrical or insulating connections) or surface science (e.g., for quick forming of surface textures or creating building blocks for more complicated formations). There are a few large groups of leftover particles, close to the middle of the electrode array which are most likely trapped due to the locally formed electric potential minimum (see Detail C in Figure 11c). The locally formed electric field minima delineated with the contour plots equipotential surfaces that encompass Detail C in Figure 11c, are most likely due to low electric field gradients (identified as large distances between the adjacent equipotential contour plots). They demonstrate the necessity of creating sufficiently strong electric field gradients in order to move the particle groups to the desired locations. If the gradient is not large enough, then, the overall force balance on the particles is insufficient to move the particles, and they remain stuck to that location until the electric field gradient is changed.

The final distribution of particles after migrating to the fourth column of the electrode array is shown in Figure 10e. A large portion, but not the majority of the particles, is located within the desired deposition region, identified with the gray dashed rectangle with the rounded corners. There are a few large groups of particles (yellow arrows in Figure 10e) and several small groups of particles (red arrows in Figure 10e) that are outside of the desired deposition region. These “straggler” groups can be explained by the local electric field minima or insufficient electric field gradient which resulted from the applied electric field distribution (see Figure 11d). These local minima and low gradient regions are located on the left side of the electrode array, between the first and third columns of the electrode array (the region with electric potential above 11 V and marked with red color). Detail E in Figure 10e shows another example of particles latching to the electrode and acting as its extension. Detail F, in Figure 10e, shows a preliminary formation of long particle deposits. These preliminary formations are similar to the smeared, small particle groups, which are identified with the red arrows in the panel. In Figure 10e, F details have similar shapes with the computed particle positions, shown as Details D and F in Figure 12d,e, respectively. They seem to be formed after the initial particle group gets stuck to the electrode and continues to chain-up with neighboring particle subgroups.

The common observation for all sets of experiments is that the particle groups do not deposit or stay in the regions with high electric field gradient and that it is possible to inadvertently trap particles in high electric potential regions where the electric field gradient is low. Additionally, stable particle groups tend to form in the low electric potential regions (<2.5 V), which corresponds to the dark blue regions of computed electric potential distribution, shown in Figure 11.

Electric potential distributions, computed with the COMSOL FEM model, are shown in Figure 11. These figures are obtained by applying an electric potential to the electrodes’ circumferences, according to the control schemes (A–D) defined in Figure 6a–d. Regions with high electric field gradients are identified with the closely spaced equipotential surfaces, whereas regions with low electric field gradients can be identified where equipotential surfaces are distanced further apart. All equipotential surfaces, denoted by the black contour plots, are spaced 0.34 V apart. The obtained computed electric potential distributions, which correspond to control schemes A–D, are shown in panels a–d in Figure 11, respectively. Detail A, in Figure 11a, indicates the region of electric potential that has a value of ~5 V, has a light blue color, and has the same arch shape as Detail A in Figure 10b. By comparing these two details, it can be seen that very large particle groups form stable deposits and are in force equilibrium with electrical forces where the electric potential difference is approximately 5 V. Detail B, in Figure 11b, shows the local electric field minima that have trapped the particle groups, as shown in Detail B in Figure 10c.

Computed test particle end positions are shown in Figure 12. Test particle end positions that correspond to the control schemes A–D are shown in Figure 12 panels b–e, respectively. Every simulation run had the same starting positions for all test particles and they are shown in Figure 12a, where 400 test particles are equally spaced between −3.5 and 3.5 mm on both x and y axes. In this figure, test particles are shown as blue dots, and the positions of the electrodes are drawn with black circles (the reader is referred to the color version of Figure 12). Detail A, in Figure 12b, shows test particle positions which correspond to the arch-like particle deposits in Detail A in Figure 10b, and to the electric potential distribution that has a 5-volt difference, as depicted in Figure 11a’s Detail A. Because no interparticle and particle–electrode interaction potential was implemented in the current FEM model, the majority of the test particles have ended up on the surface of the electrodes with the lowest electric potential (i.e., blue dots which are in contact with the black circles). These full (Figure 12b,d) or partial (Figure 12c–e) electrode surface coverings can indicate probable locations where particles would latch on the electrode during the deposition process. Comparing Figure 11a–d with Figure 12b–e shows that the test particles have migrated away from the regions with high electric potential towards the regions with low electric potential. Some particle groups have remained close to their starting positions (e.g., particle group between the first and second array column in Figure 12d, and particles between the first and third array column in Figure 12e). Their positions, when compared to the equipotential contours in Figure 11, indicate that the test particles have been stuck in the low electric field gradient regions.

To further characterize the EPμAM process, experiments presented in Figure 10 were repeated twice. In these repetitions, two parameters were varied, particle concentration was decreased from 2% to 0.2% wt., and the total applied electric current was dropped from 1 A to 0.1 A. Case A experiments used 2% wt. solution and 1 A of total electric current, case B used 0.2% wt. solution and 1 A total electric current, and case C used 0.2% wt. solution and 0.1 A total electric current.

Captured images for each case were post processed in MATLAB, and the particle group locations were extracted as a binary mask. An example of post processing is shown in Figure 13. Figure 13a–c are raw images of particle distributions at the end particle of the focusing with control scheme B. It is noticeable from the raw image data that these particle group shapes are comparable (see the surrounding area of the middle two electrodes in the third row of the electrode array and take into account the shape of 5 V electric potential region in Figure 11a, Detail A). A small exception to this observation is seen in Figure 13b, where the lower particle concentration and a lower amount of injected particle solution volume managed to only partially cover the desired deposition region in the middle of the array.

The cases were quantitively compared in the following manner: The raw image data was transformed from RGB to HSV color space, and then a binary filter mask was built to extract the particle location data. Extracted binary masks of particle locations are shown 13d-f, which correspond to the images in Figure 13a–c, respectively. Summed pixels along y- and x-axis of the images are shown in Figure 13g–i and Figure 13j–l, respectively. They depict the distribution of particles along the picture’s width and height and can be used for further analysis of particle redistribution during the experiments. Due to the sheer number of images used in all experiments, we employed a more aggregate approach, where the total binary mask area and rate of change in binary mask area were used.

The binary mask area and mask area change for performed experiments are presented as trendline plots in Figure 14. The top plot in Figure 14 shows the total number of pixels, which represent total deposition cell area covered with particles, throughout all four types of experiments, where control schemes A–D were used. To help the reader, we have marked the data from these experiments with gray and white regions in Figure 14. Data shown in Figure 13 corresponds to camera Frame 13, located at the beginning of the control scheme B region at the top of Figure 14. The bottom plot in Figure 14 shows binary mask area change, defined here as the total pixel count difference between the previous and the current camera frames.

The goal of these parameter runs was to determine the feasibility of the EPμAM process to manipulate and deposit particles when lower particle concentration and a lower electric current was used. The use of a lower particle concentration would result in material savings, and a lower electric current would allow the usage of faster and lower current-rated switching circuits. Faster switching circuits would, in turn, enable the use of a higher frequency range and expand the processing parameter space for the EPμAM process.

All three parameter runs behaved similarly when control scheme A was employed, as shown in Figure 14 top (downward-facing slope for all three cases). The focusing runs managed to decrease the particle area approximately four times, demonstrating a reliable capability to group the particles. Higher focusing areas can be achieved by increasing the number of cycles from 100 to several hundred cycles; however, this was outside of the scope for these studies.

The parameter runs with control scheme B showed some variability. The binary mask area in these experiments seemed to increase fast for the first 30 to 50 s (see trendline values for camera Frames 11 to 15 in Figure 14) and, then, started to decrease (from Frame 15 to Frame 20), at different rates for A and C cases. Cases B and C had a lower overall mask area as compared with the case A mask area. The total binary mask area for case B gradually increased throughout the control scheme B actuation. This “mask increase, then decrease” observation can be explained by the limited actuation range of each electrode in the following way: In the beginning, particles close to the actuating electrode are being moved away from it, which would increase the binary mask area. Then, after most of the particles are moved to the edges of the electrode’s effective actuation range, they are being deposited and grouped there which results in binary mask area reduction. From Figure 14 data, it seems that lower particle concentration (as in cases B and C) and in combination with the lower electric current (as in case C) causes electrodes to have lower effective range.

Experimental runs, where particles are translated to the third column via the control scheme C, have the same trends, i.e., a gradual descent of trendlines for camera Frames 23 to 34, but with different offsets for total ray mask values. The results for the case B run can be explained by the overall decreased size of the binary mask (compare mask area from Figure 13e with mask areas in Figure 13d,f), which would translate into lower total mask area during the translation experiments. The variation in the case C run is mostly due to the binary mask filter limitation to capture image regions with lower particle concentrations (smeared particle groups, indicative of lower particle concentrations, are harder to contrast with their surroundings and background scene). Another possible explanation for these variations could be due to decreased capability of the electrode array to push away low concentration particles with a lower electric current.

Experiments, where control scheme D was used to translate particles to the fourth array column, show the same trends. All three cases run have constant mask areas, with cases A and C that have similar total value, and case B with significantly lower total mask area value than cases A and C. Case B run total binary mask area can be explained with a smaller starting group of particles (compare Figure 13b to Figure 13a,c).

### 3.2. Midline Particle Deposition, a Complex Manipulation Example

A more advanced type of experiment was also employed where a set of basic actuation strategies were applied in sequence to the particle group with the goal of localized deposition between the electrode sites. The results for two types of midline depositions are shown in Figure 15.

Figure 15a–d shows the “horizontal” midline deposition of TiO_2_ particles, performed in 400 cycles. In the first 200 cycles, the particle group was squeezed between the second and third electrode array row (using control scheme E, shown in Figure 6e), followed by 100 cycles stretching the group between the first and the fourth electrode array column (see control scheme F in Figure 6f), and finishing the deposition with 100 cycles of squeezing between the second and third electrode array row (control scheme E). The initial particle droplet is shown in the first picture on the left. Subsequent images show gradual manipulation of this initial solution blob into a stretched chain of agglomerated particles spanning the whole microelectrode array. These images show that the end of the particle line agglomerated around the leftmost electrode on the third row of the electrode array, most likely the consequence of the particle group proximity to the electrode at the start of the manipulation process. Figure 15e–h shows results of the “vertical” midline deposition of the TiO_2_ particle blob. These images also show the partial attachment of the particle groups to the nearest electrode, similar to the Detail E and bottom Detail F in Figure 10e. Barring undesired latching of particle groups to the electrode surface, the complex manipulation examples demonstrate that particles can be arranged in the desired way, and away from the electrodes.

### 3.3. Midline Particle Deposit Characterization

In order to characterize the obtained particle deposits, in terms of layer thickness and deposited particles’ shape, the horizontal midline deposit experiment was performed, and a high aspect ratio deposit was formed (Figure 16). In this experiment, the initial particle blob was manipulated in the designated target region (compare panels a and e in Figure 16) with subsequent application of control schemes E and F (refer to the panels e and f in Figure 6). These control schemes were alternated four times each (eight alternations in total), and each control scheme was applied over 50 cycles during one alternation. The observed deposit formation is shown in Figure 16a–e. During the deposition process, it was observed that two small particle groups, analogous to ones shown in Figure 10 (i.e., yellow arrows), started to form in the center region of the deposit (Detail A in Figure 16d). By the end of the deposition, these two groups merged with the whole particle deposit. Close to the end of the deposition procedure, a large particle group, analogous to the large particle groups shown in Figure 10, started to form in the region near the middle leftmost electrodes (Detail B in Figure 16e). However, this particle group did not latch to these electrodes and it remained in their vicinity. The overall particle deposit (Figure 16e) was comparable in size and shape to the one shown in Figure 15d. The reconstructed video of this midline deposit formation is provided for the reader’s reference, as Appendix A.

After the deposit was formed, the excess water (~2.5 mL) was gently removed with a syringe from the deposition cell. The remaining solution was left over to evaporate and dry. it took around 3 h for the whole cell to dry completely, under normal laboratory conditions. Then, the dried particle deposit was inspected on the Olympus OLS5000-EAF laser confocal microscope. The images in Figure 17 were obtained with a LMPLFLN50xLEXT lens (1× zoom, 0.4 scan pitch, 50 brightness, 100 laser intensity, shading, and optical noise correction enabled set parameters). The resulting image’s size is 2423 × 12,121 pixels and was obtained with the stitching function (32 × 6 individual region scans) of the microscope’s control application.

The 3D scan in Figure 17a corresponds to the particle deposit shown in Figure 16e. The sample scan had an approximate area of 10.5 mm^2^ (7 × 1.5 mm), and the obtained particle deposit had a 1:5 (width-to-length) aspect ratio, in contrast to ~1:20 aspect ratio of the deposit, as shown in Figure 16e. Another notable observation was the observed bending of the deposit shape, most likely due to suboptimal drying conditions, which caused partial movement of the deposit over the glass surface. Preliminary inspection of the scanned sample identified a few features (Details A and B in Figure 17). Detail(s) A on this figure correspond to the deposits of nondispersed particle groups, deposited as a lump, which were observed with the microscope scan as a surface feature with large height. The second observation, best seen in the 2D sample intensity image in Figure 17b is a particle deposit formed due to particle collection on the edge of the evaporating water droplet. The particles deposited in Detail B came from nondeposited particles that were freely floating in the vicinity of the deposited sample. These freely deposited particles were hard to detect on the microscope 3D height scan (compare Detail B in Figure 17a with Detail B in Figure 17b). The color bars in Figure 17 have a minimum as a negative value (−0.25). This negative value originated from the tilt correction algorithm used in post processing of the scanned sample and not from the perceived negative depth. The glass substrate in the deposition cell was inspected and cleared for scratches and defects before the deposition experiment was performed.

The next step in the analysis of the deposited sample was the computation of areal roughness parameters within the OLS5000-EAF laser confocal microscope’s application, which are presented in Table 2. The reader is referred to the technical manual for the details of the used parameters [17]. The initial parameter computation was performed over the whole sample area, after which the computation was repeated over the region of interest (ROI) (see Figure 17c) under the same conditions. The obtained results for the whole sample and the ROI areas are given in the second and the third column of Table 2.

After areal parameters computation, several profile measurements were performed. In total, two vertical profile height measurements (named V1 and V2) and six horizontal profile height measurements (named H1, H2, H3, H4, H5, H6) were taken. The location of the selected profile lines is shown in Figure 18. The dashed lines’ arrows indicate the scanning direction of the performed profile height measurements, shown in Figure 19, Figure 20, Figure 21, Figure 22, Figure 23, Figure 24, Figure 25 and Figure 26. In all these figures (Figure 19, Figure 20, Figure 21, Figure 22, Figure 23, Figure 24, Figure 25 and Figure 26), the gray lines represent the data points obtained from the profile measurement, and the black lines were obtained by applying a Gaussian filter with 100 and 50 bins for vertical and horizontal data profiles, respectively. The black line plots were used for better estimation of the observed particle deposit distributions.

Figure 19 shows the obtained profile height for the V1 scan line. This profile covers the majority of the particle deposit’s length, where most of the detected height particle height is between 0.3 and 1 µm. The observed profile is rather nonuniform, with three specific distributions (0 to 500, 500 to 3500, and 3500 to 6000 µm over the profile’s length), which correspond to the three distinct particle groups shown in Figure 18 (e.g., the group near the left edge of the image, the group between H2 and H3 lines, and the group between the H5 and H6 scan lines).

Figure 20 presents the height profile of the second vertical scan line (V2). This profile does not capture a lot of deposited particles. Three particle distributions are discernible from the plot (0 to 1500, 4500 to 6000, and 6000 to 6750 µm over the profile’s length). In the first two distributions, the majority of the profile height is below 1 µm height mark, and in the case of the third distribution, which is partially overlapped with the second, the profile height rises up to 2 µm.

Horizontal height profile 1 (H1) is shown in Figure 21. The majority of the profile’s height is below 0.5 µm, with only one discernible particle distribution, spanning 200 to 1000 µm along the profile length. This profile shows the distribution of sparsely deposited particles, without clear deposit formation.

Figure 22 shows the H2 profile scan. In this figure, a single particle distribution can be detected, starting around 250 µm and ending at 800 µm on the profile length axis. The particles deposited along this line form a deposit with a ~300 µm width and an average height of around 0.5 µm.

Figure 23 depicts the height profile scan for the H3 line. In this figure, only one particle distribution is present that spans from 200 µm to 800 µm along the profile length axis. Noticeable waviness in the distribution is indicative of loose particle grouping in this region, which corresponds to the particle deposit created from the particle groups denoted as Detail(s) A in Figure 16. The average height of the profile line, within this distribution, is around 0.4 µm.

The horizontal profile scan H4 height profile is depicted in Figure 24. There seems not to be many particles deposited along this scan line, only two features can be seen on this figure. The first feature is the particle distribution spanning 400 to 600 µm profile length range, which corresponds to the small particle group deposit with an average height of around 0.250 µm. The second feature is a peak near the 975 µm length coordinate. This corresponds to the deposit height of the particle groups shown in Detail B in Figure 17, which stems from the particle grouping on the edge of the water droplet during the droplet’s evaporation.

Figure 25 shows the height profile for the horizontal scan H5 line. A single, nonuniform distribution is noticeable between 200 and 1000 µm profile length range. The average height of the distribution profile, 0.5 µm, in this scan is larger than the others which is a possible indicator of a multilayered deposit. An apparent dip, near 700 µm, corresponds to the crack in the particle deposit, most likely due to the nonuniform drying of the particle deposit in this location.

The horizontal profile for the H6 scan line is shown in Figure 26. In this profile, a single distribution of particles is seen, spanning from 400 to 1100 µm on the profile length axis. The average profile height in this distribution is around 0.5 µm.

## 4. Discussion

Our results demonstrate the feasibility of using the EPμAM process to manipulate and deposit particle groups from colloidal solutions. Simple particle group manipulation experiments demonstrate the EPμAM’s testbed capability to move and arrange particles in the desired way. Feasible strategies for dealing with “left-over” or “straggler” particle groups that did not end up in the desired deposition region include (i) increasing the number of the applied deposition cycles or (ii) changing process parameters to obtain higher electric field gradients. Undesired particle latching to the electrodes can be minimized by increasing the applied electric field frequency or implementing faster control loop operation (up to the physical capabilities of the controller).

The simple particle group manipulation experiments, shown in Figure 10, managed to order and reorder particles faster, by order of magnitude, than patterned microelectrodes [18] and via microtransfer molding [19] variants of EPD processes which operate on the order of hours. Deposition times were on a similar order of magnitude as light-based EDP variant examples, where deposition regions were defined with spatial light manipulators [20] or with laser-cut aluminum masks [21].

Some particle groups during basic manipulation experiments were not affected by the applied electric field. We suspect this is unavoidable but partially addressable by (i) stabilizing colloidal solutions to prevent premature agglomeration of the particles, (ii) calibrating the electric field strength to prevent particle overshot and sticking to the electrode surfaces. It is interesting to point out that the formed particle group in the experiments (Figure 10b, Detail A), specifically the bottom part of the group in the focusing region, has a very similar shape with the computed electric potential distribution, shown in Figure 11a, Detail A. This is most likely the result of the force equilibrium between the applied electric fields and the particle diffusion efforts. Similar results were obtained in the study of DEP-based shaping of giant unilamellar vesicles and particles [22] with octopole electrode configuration where DEP forces shaped the small oil blobs and particle groups. Both ours and Korlach’s studies used electric field modulation and obtained similar results with different material types, i.e., solid nanoparticles and unilamellar vesicles, indicating the suitability of using electric fields to shape soft and particulate matter arbitrarily. In our study case, the formed particle groups have retained their final shape after stopping the application of the electric fields. Developing more accurate FEM models for the EPμAM process, where computed electric equipotential surfaces are calibrated and validated with the used particle-fluid system, will enable relatively simple and straightforward models for numerical prediction of the final particle deposit configuration based on the computed equipotential surfaces.

Another result of the basic manipulation experiment is EPμAM’s process capability to convey the particle group at the desired location, thus, removing the requirement for exact injection of colloidal solution at the desired location; even in the case of erroneous deposition, the electric fields can be inverted, and particle group can be redeployed. There is a limit to this capability. Particle group control near the microelectrode array edges is severely limited, and after a certain particle packing ratio is achieved, the electric fields have significantly less capability to disperse tightly grouped particles. A more in-depth study of this self-assembly behavior is warranted.

The complex particle manipulation experiments demonstrate the advanced capability of the EPμAM process to order particles. Figure 15 shows particle depositions between the electrodes, unlike the depositions with the electrophoretic deposition process, where the material is being deposited on one of the electrodes, barring the use of semipermeable membranes [23] or use of deposition masks [21]. The use of electric fields to push particles in the desired location, instead of attracting them to the electrode sites, allows for deposition between the microelectrode sites, thus, essentially decoupling the maximum achievable particle deposit resolution from the number and density of electrodes in the microelectrode array. The main limitation of this push-based approach is the inverse square relationship between the strength of the electric field and the interelectrode distance that allows for the manipulation of particles only in the vicinity of the electrodes. Further studies are necessary to find an economical and practical number and density of electrodes in the microelectrode array for a given particle deposition or ordering application.

The particle deposit characterization experiment, reported in Section 3.3, detected the lack of particle layer deposit stability under the fast-drying conditions. The initial particle deposit ratio (width/length) of 1:20 degraded down to 1:5, mostly due to particle diffusion across the deposit length. Very little or no diffusion occurred on the edges of the deposited line, where larger and apparently more stable particle deposits were formed. Another reason for this deposit degradation could be due to loosely deposited particle groups (Detail A in Figure 16), which formed this part of the deposit. However, it should be noted that this deposit was formed under comparable process conditions to the ones reported in Section 3.1 and Section 3.2, where the maximum number of cycles was artificially set to a lower level. It is plausible that a more compact, and stable, deposit can be formed by increasing the number of cycles for the applied control schemes. This should be addressed in one of the next research directions. This publication’s focus and scope are on the initial deposition phase of particle deposition with electric fields, and without any post-processing steps. As previously mentioned in [6], ensuring a colloidal deposit’s post-process stability remains one of the important milestones to be achieved.

The 3D scan of the obtained midline deposit was post processed with the Olympus’s microscope computer application, where areal roughness parameters were computed, in accordance with ISO25178-2:2012 and ISO25178-2:2012 standards. The roughness parameters were computed for the whole sample and the selected region of interest (see the ROI in Figure 17c). The root mean square (RMS) height parameter (Sq) was 0.360 and 0.398 µm, for the whole sample and ROI, respectively. The 10% increase in the ROI areal roughness can be explained by the larger particle presence, relative to the area, in the ROI than across the whole sample area. The range of the obtained values 0.360–0.398 µm has an adequate match with the observed height profile averages of the particle deposit distributions. The maximum peak height parameter (Sp) was bigger in the whole area than in the ROI (8.509 to 7.512 µm, respectively). This can be explained by the presence of particle lumps (Detail A in Figure 17) and large particle groups on the edges of the midline deposit, which were taken into account during parameter computation across the whole sample but not completely in the ROI area. The maximum pit depth parameter (Sv) was marginally bigger in ROI (1.261 µm) as compared with the whole sample area (1.131 µm). The most likely explanation for these values is the higher chance of having a hollowed-out group of particles, with a hole/pit between them, located in the ROI than somewhere in the whole sample. The hollowed-out groups are probably the result of noncompact particle deposition.

Maximum height parameter (Sz) values, 9.640 µm for the whole sample, and 8.773 µm for the ROI area are directly computed from the Sp and Sv parameters. The values of Sz parameters approach 10 µm mark, which is spuriously high. This is most likely due to the use of peak values (Sp, Sz) in the Sz parameter computation and its sensitivity to the contamination artifacts (i.e., lump particle deposits, Detail A in Figure 17). Arithmetical mean height (Sa) parameter values, 0.270 µm for the whole sample and 0.328 µm for the ROI area, show what the mean of the average height difference for the average plane is. This parameter is the closest inference for the obtained particle deposit layer, for a given computation area. The Sa parameter in the ROI area is 21% higher than the Sa parameter across the whole scanned sample area, indicating moderate achievement for particle deposition within the designated ROI. The 21% increase is mostly due to the location of two large and high particle group deposits that are located in the whole sample area but outside the ROI area. These results should be taken into account with the fact that the prepared sample was not completely processed (e.g., sintered, or impregnated with polymer and cured), it was characterized as intermediate, or a “green” part in the additive manufacturing sense.

The root mean square gradient parameter (Sdq) represents the mean magnitude of the inspected surface slope. The values of this parameter for the whole sample and ROI areas are 0.379 and 0.499 µm, respectively. The Sdq parameter values are bigger than the corresponding Sa parameters, which indicates steep or very steep surface slopes. This can be explained partially with the observed measurement noise on one side, or the capability of deposited particles to form particle deposit’s edges with high inclination, on the other side. The developed interfacial area ratio parameter (Sdr) signifies the rate of increase in the surface area (i.e., an increase when the particle layer is deposited). The computed ratios for the whole sample and the ROI are 7.811 and 13.283%, respectively. The ROI Sdr is almost twice as large as the whole sample Sdr. This indicates a higher level of particle deposition amount within the ROI as compared with the whole sample area and can be used as a proxy for determination of the nonuniformity deposition level within the EPμAM process.

The height profile scans of the obtained midline particle deposit have shown that the particle deposit, after the fast-drying conditions, consisted mostly of a loose monolayer of particles, with average particle distribution heights between 0.3 and 0.5 µm. More detailed results can be obtained with more precise instrumentation (i.e., SEM); however, the current particle deposit stability is insufficient for sample preparation procedures necessary to use some of these techniques.

While performing these experiments, we have observed a few limitations. First, the large variability of the injected colloid groups (compare Figure 10a and Figure 15a,e) was mostly due to manual insertion of colloids via a syringe with a noncoated needle. Developing a more advanced insertion system and improving the insertion procedure will mitigate the majority, but not all placement variability. Secondly, certain particle patches appeared partially or minimally moved with the applied electric fields. We believe the leading cause for this issue is lack of colloidal stability during the particle agglomeration process. We did not use any stabilizers or additives because we opted for a simplified material preparation procedure. For better results, colloidal solutions prepared with stabilizing agents, and more complex electrochemical properties, will be necessary. Third, in all experiments, some particles got attached to the electrodes. The most probable cause for this undesired behavior is the operation in the low-frequency range, around 0.8 Hz, which may not be sufficiently fast to apply inverted electric potential and prevent initial particle latching to the electrode. Our current experimental setup was limited because we had to use relay switches, which have a low operating frequency, for relatively high electric currents (i.e., 0.1 to 1 A). Essential steps for addressing this issue would be to decrease, as much as possible, operating electric current and to reconfigure our electric circuits to operate in the higher frequency range. Another way would be to use smaller amounts of colloidal solution or solutions with lower concentrations, however, the tradeoff, in this case, would be significantly lower mass deposition rate.

## 5. Conclusions and Future Work

We have demonstrated localized particle deposition near, and between, electrodes of the electrode array. Achieved depositions show the utility of the EPμAM process, currently under development in our lab, for studying particle deposition and self-organization under the electric field guidance and control.

Preliminary process characterization was performed where particle concentration and electric current strength were varied. The size of the deposited particle group, obtained by the image postprocessing in MATLAB, followed similar trends and, after taking the inherent variation of the initial particle group location into account, demonstrated adequate process stability for the used parameter variation.

More advanced manipulation, which differentiates this process from alternatives where particles are attracted to a specific location, demonstrated particle conveyance and deposition between the electrodes, all this achieved without any supplemental operation or material modification.

Uncured particle deposit was characterized on a laser confocal microscope where a sparse particle monolayer was observed. The fast drying of the particle sample, after the deposition process, demonstrated the particle deposit delicacy without a proper post-processing step. It is our intention to focus our immediate attention in developing a suitable solution for this challenge. Possible approaches include sintering the obtained deposit, or by injecting a binding material (i.e., polymer) in the particle deposit and curing the obtained composite part. It is important to stress that specific post-processing steps would depend on both the type of deposited material and the end-use application for the built part and that there might be multiple suitable solutions, all dependent on the final part application.

The simplified EPμAM process model was built in COMSOL multiphysics FEM software. This model was used to compute the obtained electric field distribution and to predict test particle deposition locations. The use of this numerical model with the conjunction of EPμAM setup can accelerate the particle self-assembly study by allowing faster process parameter search and help identify outlier phenomena by comparing the obtained test particle end locations with the ones deposited experimentally.

Addressing the issue of colloidal solution stability would be an important next step. Two possible approaches to this problem are (i) adding stabilizers to the particle solution that would make it more stable, at the expense of changing its electrochemical properties and subsequently complicating the developed FEM process models; and (ii) exploring the frequency process parameter space, with appropriate high frequency switching circuits, however, this approach would require use of lower current densities and probably further miniaturization of the electrode array.

Next, a preliminary deposition study of different materials and material types is warranted. These studies should include deposition of particles with highly varied sizes and creation of composite deposits or deposition of polymer/fluid-state based solutions.

## Figures and Tables

**Figure 1 micromachines-11-00226-f001:**
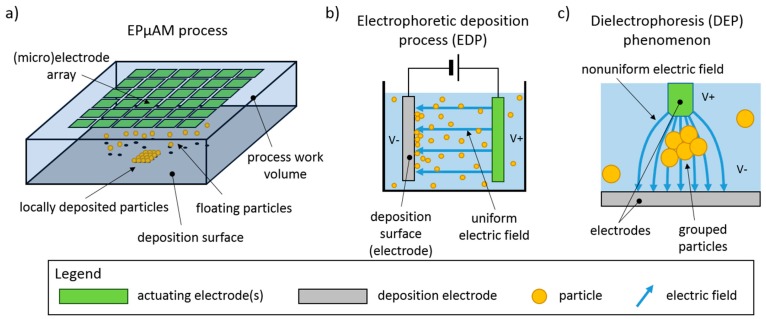
(**a**) EPμAM process schematic; (**b**) electrophoretic deposition process (EDP); and (**c**) dielectrophoresis phenomenon (DEP).

**Figure 2 micromachines-11-00226-f002:**
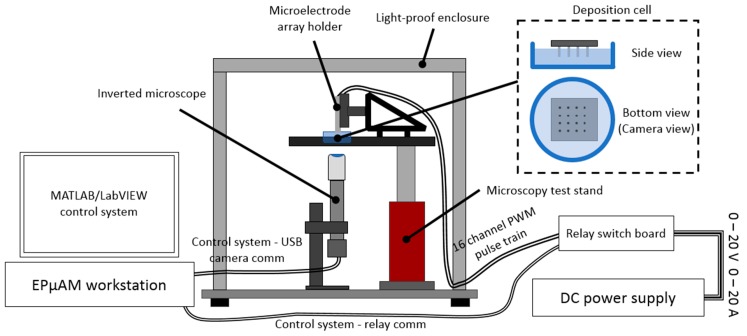
Schematic of the EPμAM experimental setup. The particle deposition cell is located within a lightproof enclosure above the inverted microscopy setup. The PC workstation is used to control the relay switchboard and to capture digital images from the microscope.

**Figure 3 micromachines-11-00226-f003:**
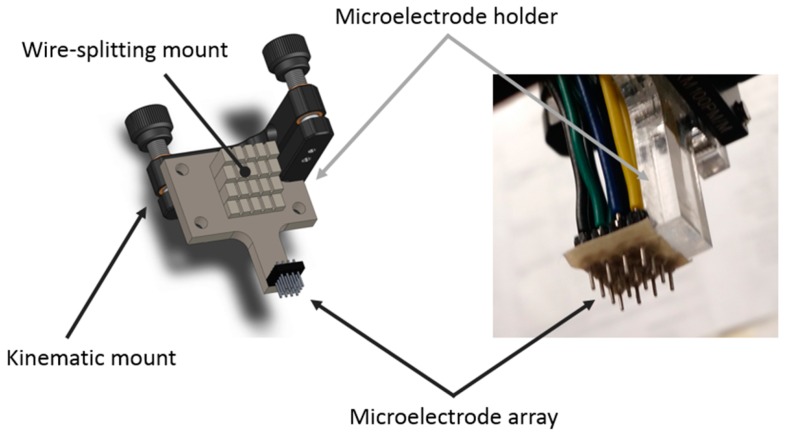
The CAD model and the fabricated microelectrode array. The array is attached to the kinematic mount (Thorlabs Kinematic Prism Mount KM100PM/M) that is used to align and level the array with respect to the bottom of the deposition cell.

**Figure 4 micromachines-11-00226-f004:**
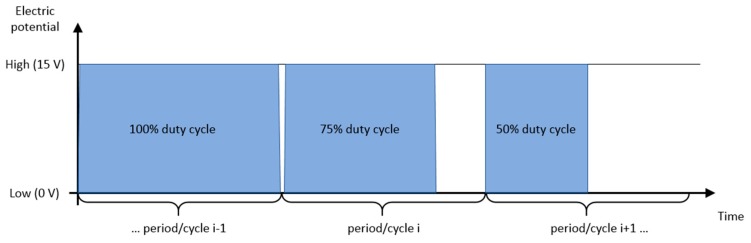
PWM duty cycle definition. The developed EPμAM control system allows individual control of each microelectrode site in the electrode array.

**Figure 5 micromachines-11-00226-f005:**
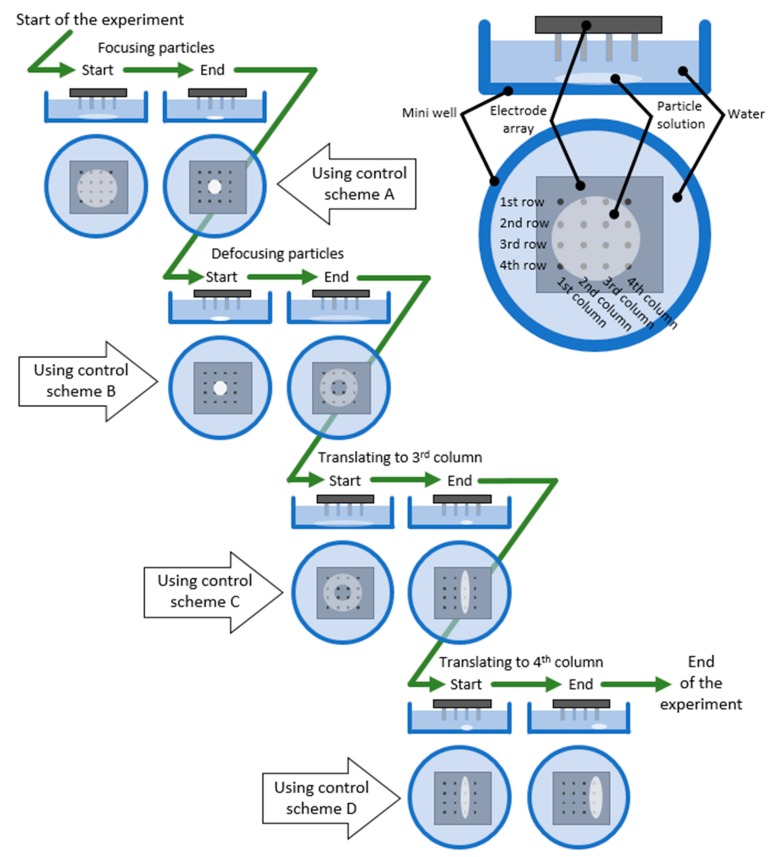
Developed workflow for the simple particle group manipulation and EPμAM process characterization experiments. The definition of commonly used terms related to the deposition cell is shown in the upper right corner of the figure. The definition of the control schemes A–D is given in Figure 6.

**Figure 6 micromachines-11-00226-f006:**
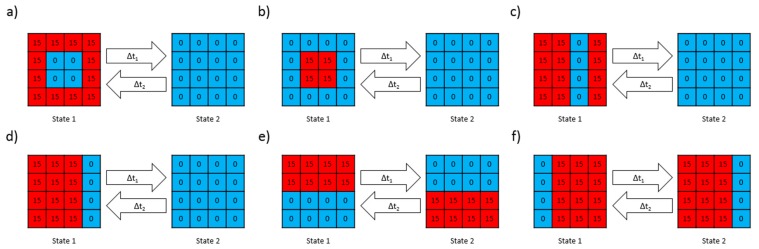
Microelectrode array actuation strategies. (**a**) The scheme for creating middle focusing potential ( control scheme A); (**b**) the scheme for the middle defocusing potential generation (control scheme B); (**c**) the translation to the 3rd column scheme (control scheme C); (**d**) the translation to the 4th column scheme (control scheme D); (**e**) the “horizontal” midline deposition (“squeezing”) scheme (control scheme E); (**f**) the “horizontal” stretching scheme (control scheme F). Each grid cell corresponds to a single electrode in the microelectrode array, as shown in Figure 3 and in the upper right corner of Figure 5.

**Figure 7 micromachines-11-00226-f007:**
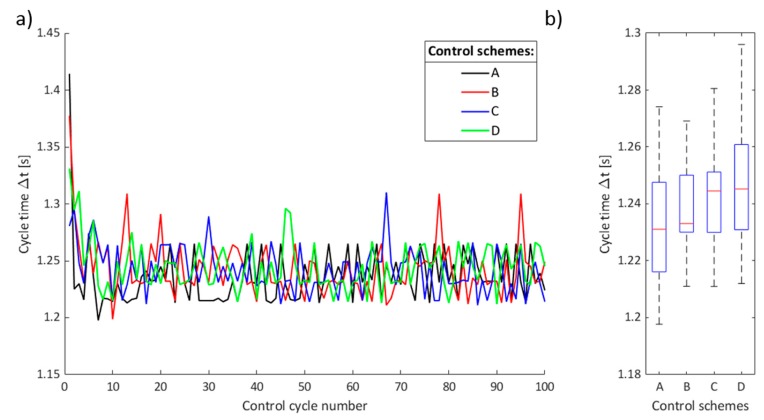
Timing results for the main control loop implementation in MATLAB. (**a**) Cycle times for A, B, C, and D control schemes; (**b**) cycle times’ boxplots for the A, B, C, and D control schemes.

**Figure 8 micromachines-11-00226-f008:**
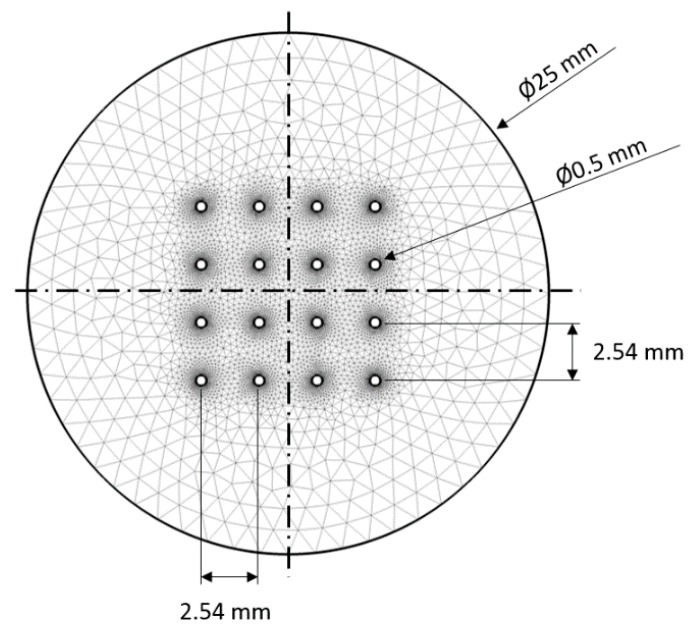
Generated FEM mesh and geometrical parameters of the 2D model. The electrode sites (small circles) are equidistant with 2.54 mm center-to-center spacing/pitch.

**Figure 9 micromachines-11-00226-f009:**
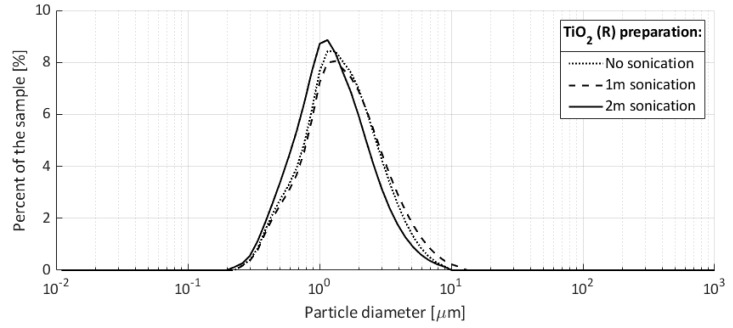
Particle size distribution for the prepared samples of rutile titania. Mean particle size for the prepared samples were 1.62, 1.78, and 1.42 μm, which corresponded to not sonicated, sonicated for 1 min, and sonicated for 2 min, respectively.

**Figure 10 micromachines-11-00226-f010:**
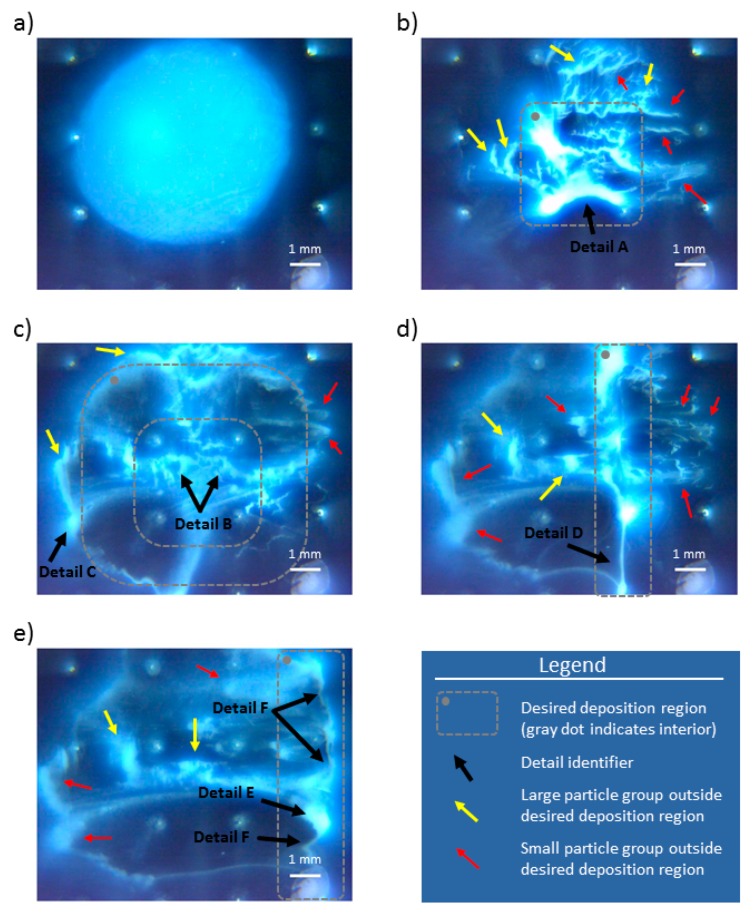
Experimental results of rutile titania (TiO_2_ (R)) particle group manipulation. (**a**) Initial particle distribution after settling on the bottom of the mini well; (**b**) particle group distribution after the particles were focused in the middle of the array with the control scheme A; (**c**) particle group spread after defocusing particles with control scheme B; (**d**) particle group arrangement after translation to the 3rd column of the electrode array with control scheme C; (**e**) grouped particles around the 4th column with control scheme D.

**Figure 11 micromachines-11-00226-f011:**
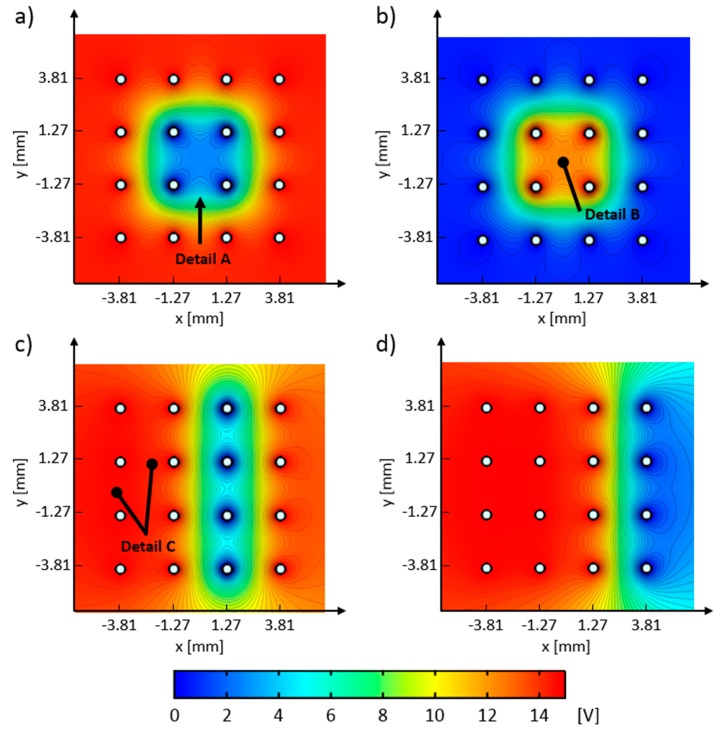
FEM computed electric potential distributions for control schemes A–D. (**a**) Computed potential distribution for focusing, control scheme A; (**b**) computed potential distribution for defocusing, control scheme B; (**c**) computed electric potential distribution for translating particles to the 3rd array column, control scheme C; (**d**) computed electric potential distribution for translating particles to the 4th array column, control scheme D. The color bar shows the range of the applied electric field from 0 V (blue) to 15 V (red). Readers are referred to the color version of the picture.

**Figure 12 micromachines-11-00226-f012:**
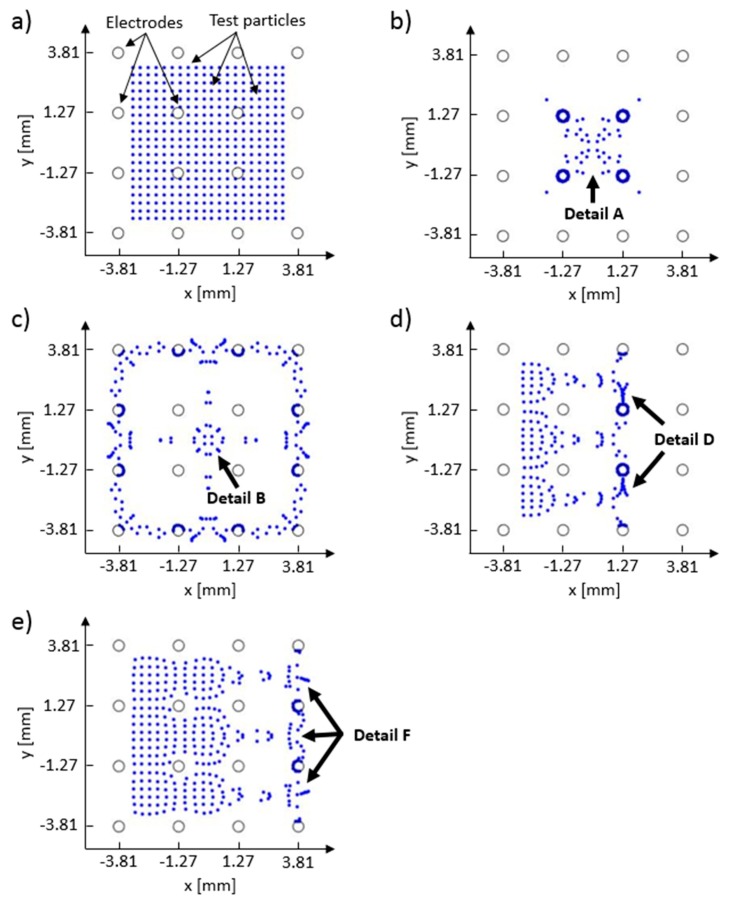
Test particle distributions in the built multiphysics 2D FEM model of the EPμAM process. (**a**) Initial test particle positions for all FEM test runs; (**b**) test particle positions at the end of the application of focusing, control scheme A; (**c**) test particle positions at the end of application of the defocusing, control scheme B; (**d**) test particle positions at the end of translation to the third array column, control scheme C; (**e**) test particle positions at the end of translation to the fourth array column, control scheme D. Large gray circles represent electrode surface positions.

**Figure 13 micromachines-11-00226-f013:**
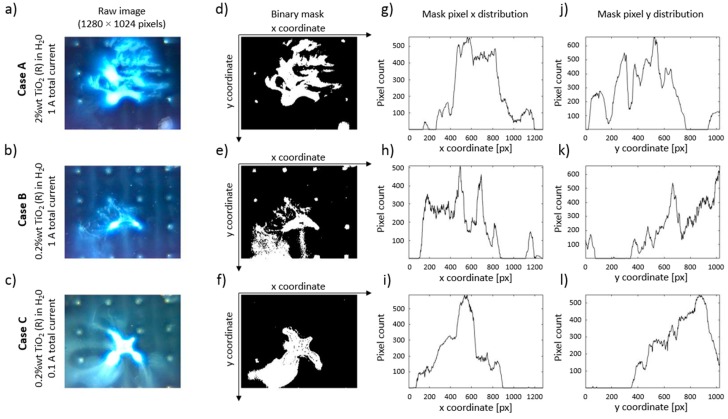
Example image post-processing results for the beginning of middle defocusing experiments. (**a**–**c**) Raw image data A, B, and C cases (panels a, b, and c, respectively); (**d**–**f**) computed binary masks for raw image data a-c, respectively; (**g**–**i**) binary mask pixel distribution (summed along image y-axis); (**j**–**l**) binary mask pixel distribution (summed along image x-axis).

**Figure 14 micromachines-11-00226-f014:**
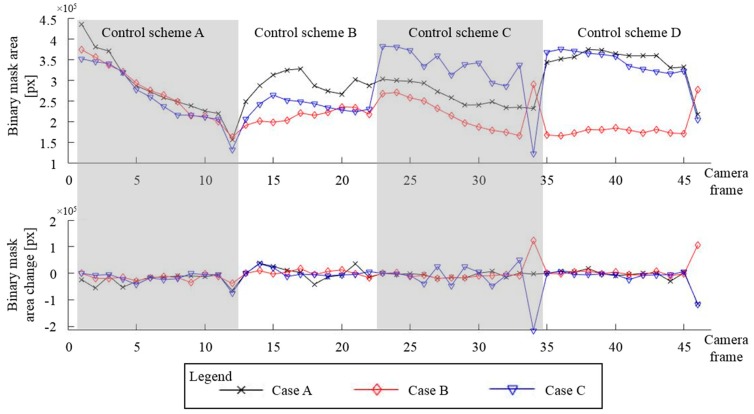
Binary masks trends (top) and their rate of change (bottom). The binary mask area is the sum of all mask pixels. The figure shows data points in the same order as the experiments were performed (middle focus, middle defocus, translate to 3rd, and translate to 4th column).

**Figure 15 micromachines-11-00226-f015:**
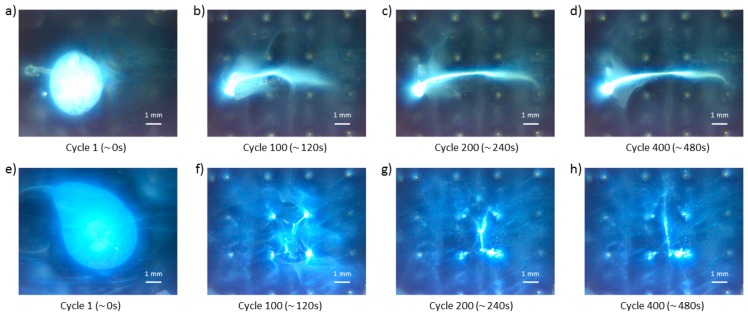
(**a**–**d**) “Horizontal” and (**e–h**) “vertical” midline depositions snapshots. The bottom right white line length is 1 mm.

**Figure 16 micromachines-11-00226-f016:**
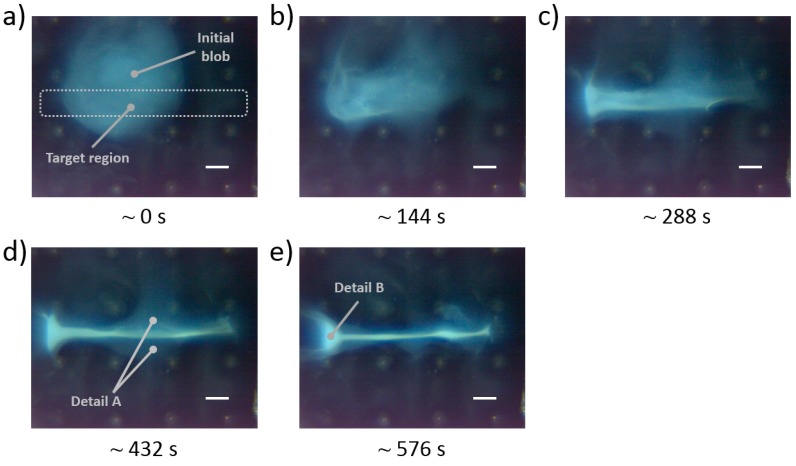
Horizontal midline deposit formation. (**a**–**e**) Image snapshots during deposit formation with control schemes E and F. The bottom time labels indicate approximate time since the start of the deposition process. The bottom right white line length is 1 mm.

**Figure 17 micromachines-11-00226-f017:**
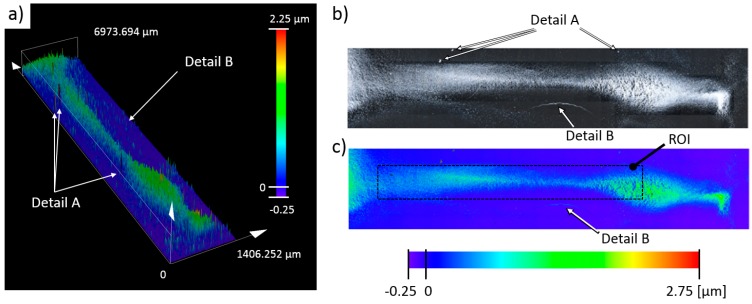
Three-dimensional (3D) scan results of the prepared sample. (**a**) 3D scan of the obtained deposit height. The color bar in the panel denotes the computed height of the whole sample; (**b**) 2D intensity image of the scanned sample; (**c**) 2D height color image of the scanned sample. ROI indicates the region of interest used for quantification of aerial roughness parameters.

**Figure 18 micromachines-11-00226-f018:**
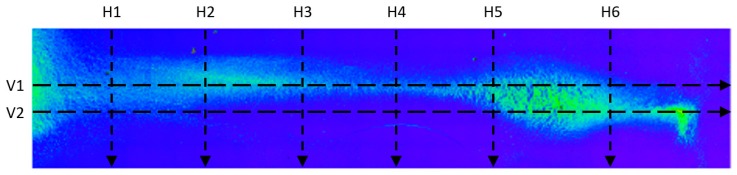
Definitions and naming convention for the performed profile scans on the deposited sample. Arrows indicate the scanning direction for the profile height lines shown in the following figures.

**Figure 19 micromachines-11-00226-f019:**
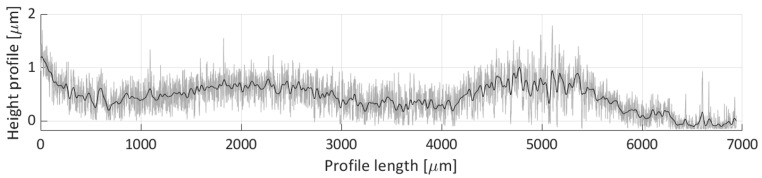
Height profile for the V1 line.

**Figure 20 micromachines-11-00226-f020:**
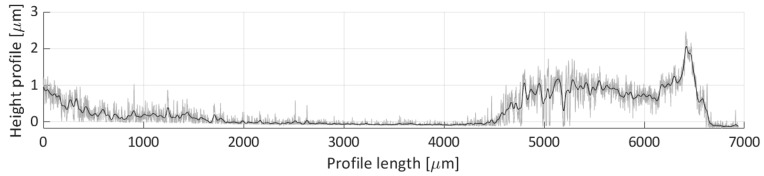
Height profile for the V2 line.

**Figure 21 micromachines-11-00226-f021:**
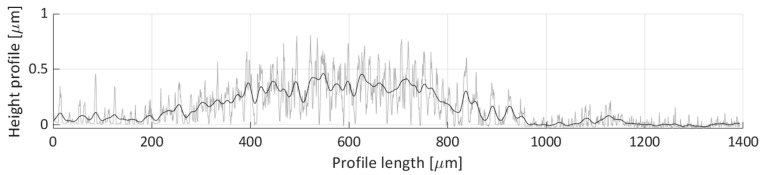
Height profile for the H1 line.

**Figure 22 micromachines-11-00226-f022:**
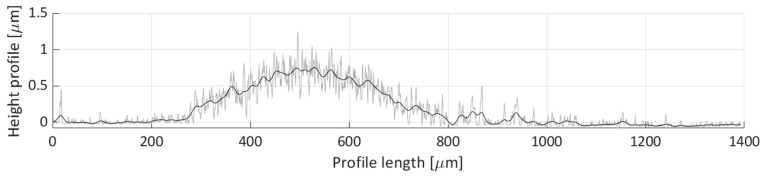
Height profile for the H2 line.

**Figure 23 micromachines-11-00226-f023:**
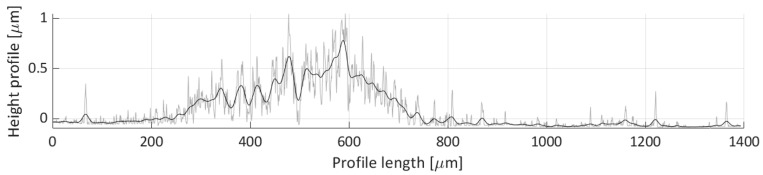
Height profile for the H3 line.

**Figure 24 micromachines-11-00226-f024:**
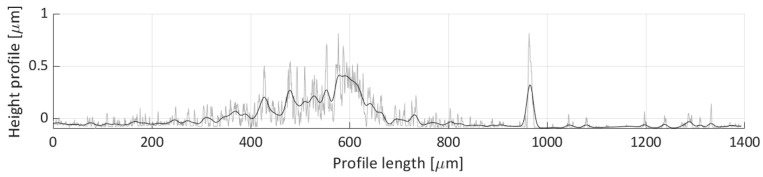
Height profile for the H4 line.

**Figure 25 micromachines-11-00226-f025:**
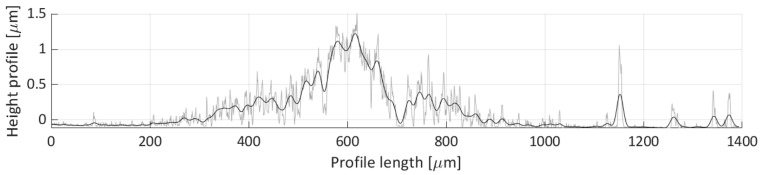
Height profile for the H5 line.

**Figure 26 micromachines-11-00226-f026:**
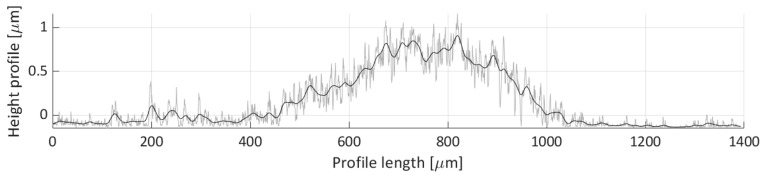
Height profile for the H6 line.

**Table 1 micromachines-11-00226-t001:** Physical and numerical model parameters (rutile titania particles and water fluid medium).

Parameter	Value	Unit
ε_r__,fluid_ (fluid relative permittivity)	80.1	1
*ρ*_fluid_ (fluid density)	1000	kg/m^3^
μ_dynamic,fluid_ (fluid dynamic viscosity)	8.9e−7	Pa·s
σ_fluid_ (fluid electrical conductivity)	0.023	S/m
Z_ref_ (model reference impedance)	50	Ω
*ρ*_particle_ (test particle density)	4260	kg/m^3^
Z (test particle charge number)	0.1	1
*d*_particle_ (test particle diameter)	3	µm
Test particle grid release coordinates	(−3.5:3.5) × (−3.5:3.5)	(mm) × (mm)
Test particle initial velocities (all)	0	m/s
Physics force type enabled	(electric, DEP, drag)	n/a
ε_r__,particle_ (test particle relative permittivity)	1000	1
σ_particle_ (test particle electrical conductivity)	1e-13	S/m
Drag force law type	Stokes law	n/a
Simulation step 1 parameter: frequency	1	Hz
Simulation step 2 parameter: time range	(0:0.5:110)	s
Step 1 solver type	MUMPS	n/a
Step 2 solver type	GMRES	n/a

**Table 2 micromachines-11-00226-t002:** Computed areal roughness parameters for the whole sample and the region of interest (ROI) areas, indicated in Figure 17c.

Areal Roughness Parameter	Parameter Description (ISO25178-2:2012 and ISO25178-2:2012)	Unit	Whole Sample Area	ROI Area
Sq	Root mean square height.	µm	0.360	0.398
Sp	Maximum peak height.	µm	8.509	7.512
Sv	Maximum pit depth.	µm	1.131	1.261
Sz	Maximum height.	µm	9.640	8.773
Sa	Arithmetical mean height.	µm	0.270	0.328
Sdq	Root mean square gradient.	n/a	0.379	0.499
Sdr	Developed interfacial area ratio.	%	7.811	13.283

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
