# Peer review of "Manipulation and Localized Deposition of Particle Groups with Modulated Electric Fields"

_micromachines, 2020, doi:10.3390/mi11020226_

Round 1
Reviewer 1 Report
In this paper, the authors reported the simulation and preliminary experiments on manipulation and localized deposition of particles using electrophoretically-guided micro additive manufacturing (EPuAM)that they called. Such manipulation processes of micro/nano particles are useful for microscale additive manufacturing. However, in the current conditions, the accuracy and resolution of their method aren’t sufficient for additive manufacturing processes in microscale. In addition, although resultant structures was observed by an optical microscope, the detail of the aggregated particles weren’t observed due to low quality of photos shown in Fig. 10 and 13. It is required to observe the localized particles with scanning electron microscope (SEM), and discuss the 3D shapes of the aggregated particles. The shape of the aggregated particles are one of the most important parameters for determine the voxel size and shape of their additive manufacturing process in the future.
I also think it is better to modify the introduction. The authors mentioned general background of particle assembly, but the motivation, originality and advantages of their method are unclear. In particular, although they mentioned DEP can manipulate particles at great precision compared to EDP in lines 67 – 70, they use EDP in their method. It is better to explain what is the advantage of their method simply. In addition, although they use needle-like electrode array in experiments, most of previous research use planer electrode pattern. What is the advantage and disadvantage of needle-like electrode array? Is it a unique point of your method? I think the originality and uniqueness should be mentioned clearly.
From the above reasons, the paper cannot be accepted for the publication in Micromachines at the current conditions.I think this paper should be resubmitted after adding experimental results of SEM observation of aggregated particles, making additional discussions and modifying the introduction.
Reviewer 2 Report
I have finished reviewing the manuscript Ref. No: micromachines-710886, Titled " Manipulation and localized deposition of particle groups with modulated electric fields. In this study. The process uses modulated electric fields to manipulate and deposit particles from colloidal solution in a contactless way and is named Electrophoretically-guided Micro Additive Manufacturing (EPμAM). The experimental and FEM results were cross-compared and analyzed; observed process limitations are discussed and followed by a comprehensive list of possible future steps. This paper can be accepted for publication.
Reviewer 3 Report
It is good article about manipulation and localized deposition of particle groups with modulated electric fields.
I have a few comments on the article.
1) In my opinion, the literature review should be strengthened by articles on the influence of external electric potentials on the properties of metals. Authors may use aluminum research papers in their analysis (Konovalov, S.V., Danilov, V.I., Zuev, L.B., Filip'ev, R.A., Gromov, V.E. On the influence of the electrical potential on the creep rate of aluminum (2007) Physics of the Solid State, 49 (8), pp. 1457-1459. DOI: 10.1134/S1063783407080094; Gromov, V.E., Ivanov, Yu.F., Stolboushkina, O.A., Konovalov, S.V. Dislocation substructure evolution on Al creep under the action of the weak electric potential (2010) Materials Science and Engineering A, 527 (3), pp. 858-861. DOI: 10.1016/j.msea.2009.10.045).
2) Authors talk about electric fields. However, the characteristics of electric fields are not given in the article. Electric potentials only indicated.
3) Authors have demonstrated localized particle deposition near, and between electrodes of the electrode array. You must specify the thickness of the formed material.
Round 2
Reviewer 1 Report
Dear Authors,
The authors modified the manuscript according to reviewers' comments. Although SEM observation is not included, confocal microscope images are added. The results help readers to understand the resultant structure of aggregated particles. In addition, the introduction is also modified to make the originality clear. I think the revised manuscript can be accepted for publication in Micromachines.
Reviewer 3 Report
I see that the authors have removed all the comments. In my opinion, the article is ready for publication in the journal.